# Predicting DNA Content Abnormalities in Barrett's Esophagus: A Weakly Supervised Learning Paradigm

**Caner Ercan** [1]                                    CErcan@mdanderson.org
**Xiaoxi Pan** [1,2]                                    XPan7@mdanderson.org
**Thomas G. Paulson** [3]                              TPaulson@fredhutch.org
**Matthew D. Stachler** [4]                          Matthew.Stachler@ucsf.edu
**Carlo C. Maley** [5]                                      Maley@asu.edu
**William M. Grady** [3,6]                              WGrady@fredhutch.org
**Yinyin Yuan** [1,2]                                  YYuan6@mdanderson.org

[1] *Department of Translational Molecular Pathology, The University of Texas MD Anderson Cancer Center, Houston, TX, USA*
[2] *Institute for Data Science in Oncology, FA1-Quantitative Pathology and Medical Imaging, The University of Texas MD Anderson Cancer Center, Houston, TX, USA*
[3] *Translational Science and Therapeutics Division, Fred Hutchinson Cancer, Seattle, WA, USA*
[4] *Department of Pathology, University of California San Francisco, San Francisco, CA, USA*
[5] *Arizona Cancer Evolution Center; Biodesign Center for Biocomputing, Security and Society; School of Life Sciences, Arizona State University, Tempe, AZ, USA*
[6] *Department of Medicine, University of Washington School of Medicine, Seattle, WA, USA*

**Editors:** Accepted for publication at MIDL 2024

## Abstract

Barrett's esophagus (BE) is the sole precursor to esophageal adenocarcinoma (EAC), and is an opportunity for developing biomarkers for cancer risk assessment. DNA content abnormalities, including aneuploidy, have been implicated in the progression to EAC in BE patients, but molecular assays require valuable tissue for its detection. We propose utilizing images from routine histology to detect ploidy status using deep learning.

Employing a weakly supervised deep learning approach, multi-instance learning (MIL), we trained a model to predict ploidy using hematoxylin and eosin-stained whole slide images of endoscopic biopsies and flow cytometry results. The study introduces a novel image augmentation method for MIL, sequentially altering features from original and augmented images during training loops. This method improved the average area under curve (AUC) from 0.43, 0.64 and 0.81 for ResNet50, DenseNet121 and REMEDIS foundation model, respectively (training without any augmentation), to 0.61, 0.87 and 0.91 with the proposed augmentation strategy.

The top-performing model, employing foundation model as the backbone, achieved 0.93 AUC and 83% balanced accuracy to predict aneuploidy in the test cohort biopsies (n=279). Across all the patients (n=123), predicted aneuploidy status was correlated with progression to EAC (p=6.55e-06), similar to correlation with ploidy status based on flow cytometry results (p=2.84e-7). Supporting the findings, histologic nuclear features typically associated with DNA content abnormalities such as enlarged and hyperchromatic nuclei were seen in the samples called abnormal compared to the control diploid samples.

In conclusion, our deep learning model efficiently predicts aneuploidy, a mechanism that has been shown to underpin BE progression to EAC. This method, preserving precious biopsy tissues, complements routine histology, offering potential for identifying individuals at high risk of progression through molecular-based advancements.

**Keywords:** Multiple instance learning, weakly supervised learning, Whole Slide Image Classification, Computational pathology, Image augmentation, clinical prediction biomarker, cancer molecular biomarker, precancer

## 1. Introduction

Due to chronic gastroesophageal reflux and the resultant inflammation, the lower esophagus can go through a metaplastic change where the normal squamous lining is replaced with columnar cells that develop intestinal-like differentiation, Barrett's esophagus (BE). BE is the sole detectable precursor lesion for esophageal adenocarcinoma (EAC), a deadly cancer with an increasing incidence (Desai et al., 2012). The progression to EAC is characterized by progressive histologic and genetic alterations of the BE (Killcoyne and Fitzgerald, 2021). DNA content abnormalities, defined as the presence of aneuploidy (hypo-, hyper- and tetraploidy) or an increased percent (>6%) of cells in the G2/M phase of the cell cycle, have been implicated in the development of EAC in BE patients (Reid et al., 1992; Rabinovitch et al., 2001; Hadjinicolaou et al., 2020). However, the existing molecular assays for DNA content abnormality detection (e.g., flow cytometry) necessitate the use of precious tissue samples, leading to difficulties in validation studies. Our objective is to develop a deep learning-based image analysis tool to predict the DNA content status from the whole slide images (WSI) of standard haematoxylin and eosin (H&E)-stained endoscopic biopsies.

Fully-supervised approaches have achieved remarkable success in detecting a wide range of objects (e.g. cell nucleus, tumor), relying on pathologists' annotations as the ground truth (Montezuma et al., 2023). However, when dealing with tasks that only have WSI-level annotations and lack local features identifiable by pathologists, such as molecular alterations, this approach faces limitations. To address this problem, multi-instance learning (MIL) emerges as a viable solution (Ilse et al., 2018). In this framework, the dataset is composed of bags (in here WSIs) with classification labels, where each bag contains multiple instances (patches of a WSI) without labels. If a bag contains at least one positive instance, it is labeled as positive; otherwise, it is labeled as negative.

In recent years, many researchers have implemented MIL methods in computational pathology using convolutional neural networks (Wang et al., 2019). Lu et al. (2021) proposed Clustering-constrained Attention Multiple Instance Learning (CLAM), which derives from the attention-based MIL framework introduced by Ilse et al. (2018). This pipeline uses an attention mechanism to identify critical regions associated with diagnosis automatically, and then generates global features based on this region. However, there is currently a scarcity of research applied to BE biopsies using this approach.

Data augmentation is effective in tackling a lack of generalisation, and data memorization. In computational pathology, augmentation can be done at the image level and data level. Image-level augmentations, such as bluring, zoom, and color brightness change, can serve dual purposes, increasing resilience of model performance to image quality alterations and increasing training data. (Krizhevsky et al., 2012) However, this traditional approach is not directly applicable to MIL training. On the other hand, the data-level augmentation approaches working on increasing the training data on feature vector (embedding) or bag (WSI) level data mixtures and pseudo-data generations.(Liu et al., 2023; Gadermayr et al., 2023; Yang et al., 2022) Contrary to most training approaches that involve encoding images to low-level features and training the model on-the-fly, MIL takes a different path by aggregating patch features and using them sequentially for training the network. This difference makes the traditional image augmentation method, application of random transformations during each epoch, impractical. To address this challenge, we introduce an innovative method. Multiple sets of low-level patch features are generated, with the initial

set derived from original images and subsequent sets incorporating random image transformations before feature extraction. During MIL training, a set of augmented features is employed every other epoch. This approach ensures that the model is primarily trained with original images while harnessing the advantages of image augmentation to improve overall performance and generality. Additionally, we integrated Pseudo-Bag Mixup Augmentation (PseMix) into our pipeline to examine the impact of feature-level data augmentation (Liu et al., 2023). PseMix is an effective Mixup variant for WSI classification. In this method, instances from WSI bags are first clustered into different phenotypes, and then, each bag undergoes phenotype-stratified sampling. Comparative experiments and ablation studies confirm that PseMix is an effective Mixup variant for WSI classification (Liu et al., 2023).

In this study, our objective was to develop a deep learning pipeline utilizing the MIL paradigm for predicting aneuploidy in BE. Our key contributions include: (1) To the best of our knowledge, this is the first study applying deep learning for predicting DNA content status based on histology images of BE. For this aim, we utilized a unique BE biopsy dataset having known DNA content results. (2) We utilized the MIL framework in the context of BE to predict cancer outcome-related features. Additionally, we examined the changes in BE epithelial cell features within regions exhibiting high aneuploidy probability to elucidate the connection between MIL model outputs and histology. (3) The study introduces a novel augmentation method for histology images in BE prediction.

## 2. Methods

### 2.1. Dataset

H&E-stained WSIs of endoscopic biopsies from BE were utilized in this study. The biopsy slides, sourced from the Seattle Barrett's Esophagus Annotated Resource (BEAR), constituted the training and validation sets. All research participants contributing clinical data and biospecimens provided written informed consent, governed by oversight from the Fred Hutchinson Cancer Center IRB Committee D (reg. ID 5619).
Biopsies obtained from patients were divided into two halves. One half underwent processing for flow cytometric DNA content assessment, as described previously (Reid et al., 1992). The other half was subjected to standard fixation and staining with H&E. The H&E-stained slides in the first cohort were scanned at 20x (0.4548 µm/pixel) using a Hamamatsu NanoZoomer. An independent cohort, composed of patients from a different subset of BEAR, served as the test set, with H&E-stained slides scanned on an Leica Aperio AT2 slide scanner at 20x (0.5013 µm/pixel). (Figure 1A) Slides with significant out-of-focus issues, air bubbles, and tissues with aneuploid results lacking BE regions were excluded from the study. The flow-analyzed tissue pieces were manually selected using the rectangular regions of interest, generating two sets for each slide when possible. The first cohort consisted of 388 slides (51 aneuploid, 337 diploid) from 34 patients, while the second cohort had 279 slides (36 aneuploid, 243 diploid) from 89 patients. The first cohort (training) underwent Monte Carlo cross-validation, being split into training (85%) and validation (15%) datasets constrained to the patients, while the second cohort served as the test dataset.

### 2.2. Aneuploid prediction from tissue space

We adapted the CLAM framework to train a deep neural network classifier for WSI-level, predicting DNA content status as abnormal or normal (Stachler et al., 2015)

**Preprocessing and Feature extraction:** The tissue segmentation and patching tasks were achieved as described previously (Lu et al., 2021). After segmentation, each tissue region was split into patches with a size of 256 x 256 pixels without downsampling to fit the input capacity of deep learning models. (Figure 1B)

Following patching, convolutional neural networks were employed to extract feature vectors from each image patch in every WSI. To assess performance differences, differences, we utilized ResNet50 (He et al., 2016) and Densenet121 (Huang et al., 2017) models pre-trained on ImageNet, along with the REMEDIS 152x2 small foundation model (Azizi et al., 2023, 2022), pre-trained on The Cancer Genome Atlas. Each network underwent modification to convert its output into a 1,024-dimensional feature vector for every patch.

**Augmentations and Training:** To implement image augmentation, four combinations of image transformations, randomly applied with different probabilities, were employed on all patches. (Appendix A) This ensured diversity in the transformations applied across the dataset. Each transformation was assigned a probability and factor, resulting in four additional sets of features for analysis. Additionally, a consistent seed was used for each set of augmented features to maintain uniformity across different backbones. (Figure 1C)

The training iterations began with using the feature vectors which were computed using the original image patches. (Figure 1D) For every other epoch, we used one of the feature vector sets which were extracted from the images after image augmentations. This strategy uses raw and augmented features simultaneously, expanding the feature pool and making the learning process effective. For the data augmentation experiment, PseMix data augmentation (Liu et al., 2023) was applied during training sessions. Training sessions were

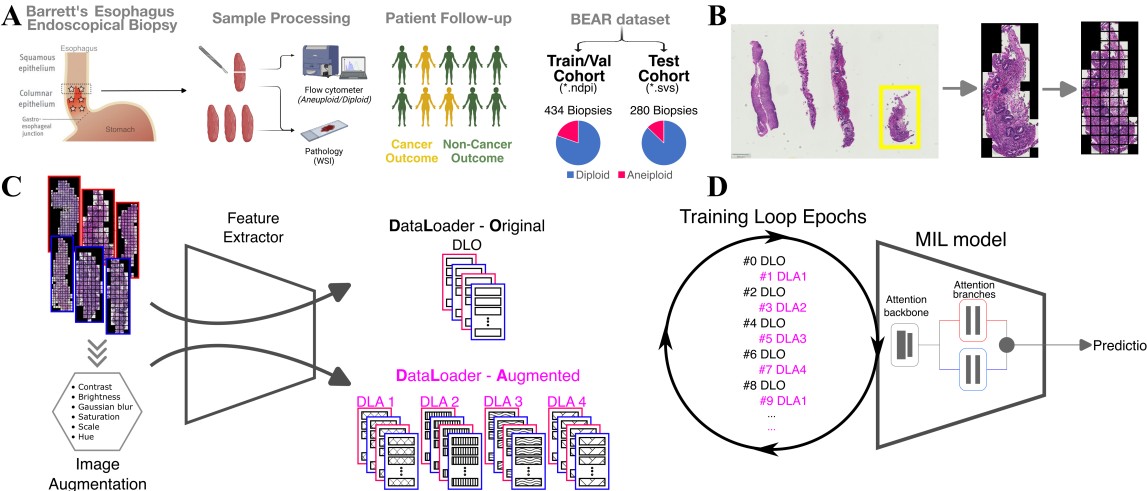

Figure 1: **Sample Collection and Model Training (A)** Unique BEAR dataset was utilized. Training and test dataset slides were scanned in different institutes using different scanners. **(B)** Image patches were extracted from tissue regions analyzed by flow cytometry. **(C)** Feature embeddings were extracted using pretrained backbone models. For the image-based augmentation seperate dataloaders were generated from original (DLO) and augmented (DLA) images. **(D)** Alternation between original and transformed image-based dataloaders during training loops.

initiated with an initial learning rate of 1e-3 and a scheduler with a 0.5 factor, 5 epochs patience, and an 8-epoch cooldown. We employed the Adam optimizer (Kingma and Ba, 2017), with a 1e-5 weight decay, and dropout with a 0.25 probability. Focal cross entropy was used as loss function. Model size was set to small and bag weight was set as 0.7. The training sessions were conducted with a minimum of 60 epochs, and an early stopping mechanism with 20 epochs patience. For the WSI-level classification, the optimum threshold for probability was calculated on the training dataset using the ROC curve based on the lowest $TPR - (1 - FPR)$ level.

### 2.3. Cell Nucleus Segmentations & Classification

For the investigation of aneuploidy histological features, the cell features were analysed and compared between top-attention scored patches. For computation of cell nucleus features, we trained a nucleus instance segmentation and classification model with our own data using StarDist algorithm. (Weigert et al., 2020) Fifteen WSI's from the test cohort slides were selected randomly and the cell nuclei in selected regions were annotated by one pathologist (C.E.). For the training, 14,576 cells (7464 immune cells, and 7112 epithelial cells) were annotated. After the inference, QuPath (v.0.4.4) (Bankhead et al., 2017) was used for calculation of nuclei intensity (n=5), texture (n=13) and shape (n=6) features.

### 2.4. Statistics

For statistical comparisons for the cell feature analysis, Student's t-Test was used. For cancer outcome analysis, Pearson's Chi-squared test was used. Confusion matrices were obtained using cvms (Olsen et al., 2023). All statistical tests were two-sided, and $P < 0.05$ was considered statistically significant. To adjust P values for multiple comparisons, the Benjamini–Hochberg method was used. Plotting was done using ggplot2 v.3.4.4 and ggpubr v.0.6.0 packages. All statistical analyses were conducted in R v.4.3.1, except for ROC analysis, which was performed using scikit-learn v.1.3.2 in Python v.3.11.6. Symbols used in figures represent following; ns : not significant, p-value <0.05:*, <0.01:**, <0.001:***

## 3. Results

### 3.1. Effective Aneuploidy Prediction with DLO/DLA Alternation

The MIL model was trained using the flow-cytometry results (DNA content abnormal or not) as the slide level annotation and BE biopsy slide images using CLAM framework. Feature maps for each WSI were generated using three different backbones: ResNet50, DenseNet121, and the Foundation Model (REMEDIS). Alongside the feature maps generated from original images, four additional sets were created after applying various combinations of image augmentation functions and factors. (Figure 1C)

Training involved the combination of a dataloader based on original images (DLO) and four dataloaders based on different settings of augmented images (DLAs). (Figure 1D)

Table 1: The Performance of Different Backbones

|  | Validation Dataset | | Test Dataset | |
|  | AUC | Balanced Accuracy | AUC | Balanced Accuracy |
| --- | --- | --- | --- | --- |
| ResNet50 | 0.741 | 0.722 | 0.723 | 0.668 |
| DenseNet121 | 0.778 | 0.733 | 0.877 | 0.50 |
| Remedis 152x2 | **0.82** | **0.75** | **0.93** | **0.831** |

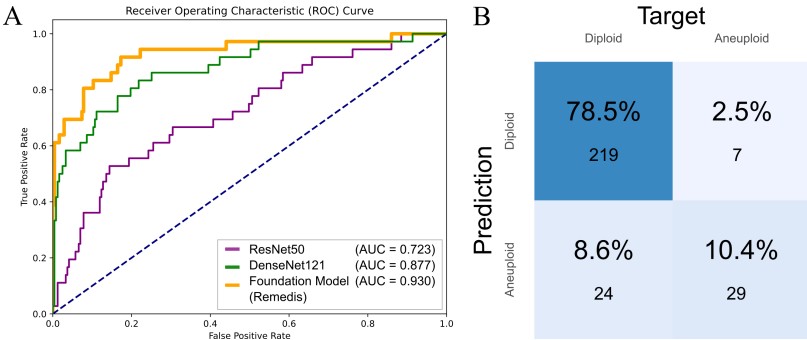

Figure 2: **Performance of image-based augmentation applied models: (A)** The utilization of foundation model as the backbone achieved 0.93 AUC and performed better compared to ResNet50 and DenseNet121, 0.72 and 0.88 respectively. **(B)** The balanced accuracy of foundation model based training was 83%

During the training, the dataloaders were alternated between the DLO and one of the DLAs for each epoch in a sequential manner. This approach ensured that the features based on original images had a primary impact on training, while incorporating the effects of various image transformations into the training process. All training sessions utilized five training/validation splits.

The proposed augmentation strategy, DLO/DLA alternating training loops combined with the foundation model as the backbone, achieved the highest area under the curve (AUC) score of 0.93 (Table 1). In comparison, ResNet50 and DenseNet121 achieved AUCs of 0.72 and 0.88, respectively (Figure 2A). The balanced accuracy of the foundation model-based DLO/DLA alternating model reached 0.83 (Figure 2B). Without augmentation, the average AUC from original images was 0.80, 0.43, and 0.81 for ResNet50, DenseNet121, and REMEDIS, respectively. The new augmentation strategy improved the performance of DenseNet121 and REMEDIS, with average AUCs of 0.61 and 0.91, respectively (Figure 2A). In the ablation study, the proposed method was found to be superior to other settings (Appendix B-C).

In the experiment evaluating PseMix feature-level data augmentation in a single iteration, the augmentation method produced similar performance to training without augmentation for DenseNet121 and the foundation model. However, when combined with proposed image augmentation, the AUC improved from 0.53 and 0.48 to 0.64 and 0.76, respectively (Appendix D). In contrast, training sessions using only image-based augmentation showed superior performance, with AUCs of 0.88 and 0.93, respectively. Training sessions using features from ResNet50 yielded the best performance in the no augmentation setting. Interestingly, training sessions with PseMix resulted in significantly inferior performance.

### 3.2. Decoding Aneuploidy: Cancer Outcome and Morphology Variances

To assess the predictive relevance of our model, we compared cancer outcomes (CO) between patients with and without DNA content abnormalities. Based on both flow results and our model's predictions, patients with abnormalities have significant higher rates of CO (p-value 2.84e-07 and 6.55e-06, respectively) (Figure 3A). 55.6% (10/18) of patients with CO had aneuploidy based on flow results while 66.7% (12/18) based on predictions. This result shows the model predictions can be informative for patient clinical outcome as well.

Finally, we aimed to investigate aneuploidy-specific histological features using the MIL model predictions. This investigation serves two purposes: understanding relevant features for aneuploidy predictions by the MIL model and validating these features with known aneuploidy-specific features. In analyzing histological features, we compared cell features between top-attention scored patches from each WSI. This analysis provided valuable insights into the relevant features for DNA content classifications. Notably, patches categorized as abnormal exhibited enlarged, hyperchromatic nuclei with a loss of cell nucleus orientation and frequent mitotic figures (Figure 3B). In contrast, patches from diploid samples often displayed no signs of atypia, with rare reactive changes.

Furthermore, we calculated and compared cell nuclei features in the top attention-scored patches between the two groups. For this purpose, we trained a cell nucleus instance segmentation and an immune cell-epithelial cell classification model and achieved 76.7% accuracy and 86.8% F1-score (Table 1, Figure 3C). Cell feature comparison analysis revealed significant differences between diploid and abnormal epithelial cells. Abnormal epithelial cells exhibited notably increased cell nucleus size compared to diploid ones. Additionally, we conducted a texture analysis, uncovering significant differences in mean nucleus entropy, standard deviation of inverse difference moment, and difference variance values between the top-attention-scored/accurately predicted diploid and abnormal groups. (Figure 3D)

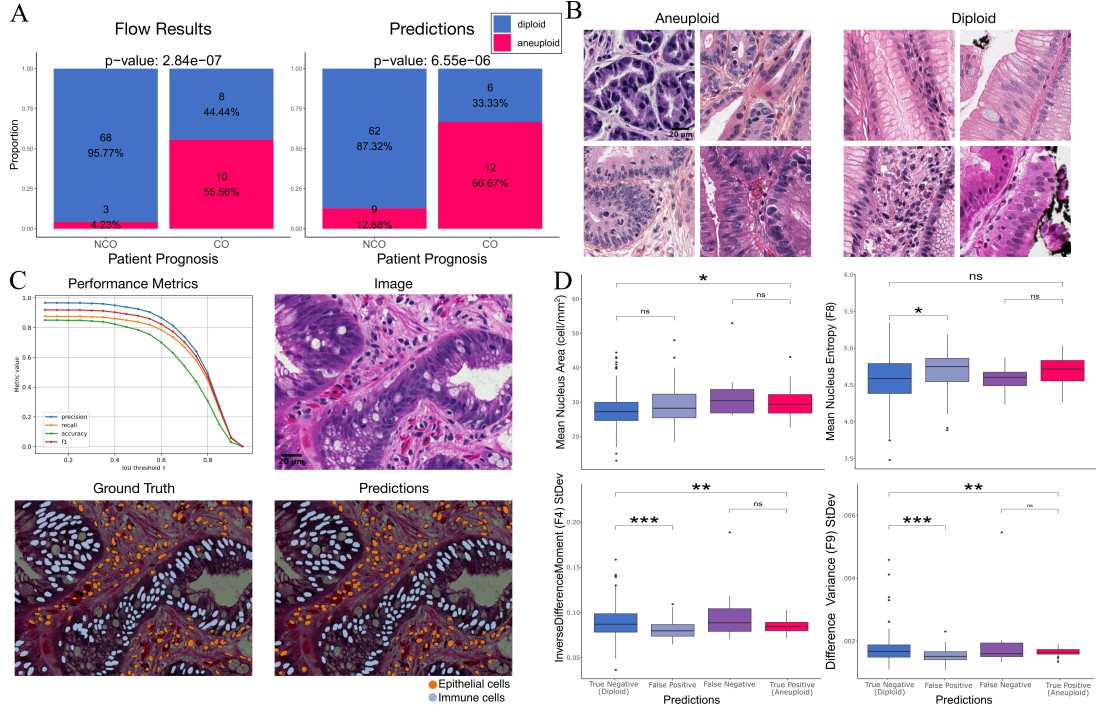

Figure 3: **Histological and Clinical Correlations: (A)** Patients with aneuploidy have significantly higher rates of cancer outcome. **(B)** Patches with aneuploidy exhibited enlarged, hyperchromatic nuclei and frequent mitotic figures. **(C)** Nucleus segmentation and classification model was trained for further investigation. **(D)** Aneuploid cells exhibited significantly increased nucleus size and altered texture.

## 4. Discussion

In this study, we developed an image-analysis model capable of predicting DNA content abnormalities on H&E stained BE biopsy WSIs, using MIL as a weakly supervised training framework. Our findings revealed a correlation between DNA content predictions and cancer progression, a well-known aspect of BE. The nature of image analysis, which allows preservation of precious small tissue samples, positions it as a promising candidate for application in routine clinical practice. Despite limited efforts in this direction (Yu et al., 2024), to the best of our knowledge, this study marks the first application of deep learning to predict DNA content status based on histology images of BE.

MIL approaches allow us to stratify patients using slide-level labels in a weakly supervised setting with exceptional clinical-grade performance (Campanella et al., 2019; Lu et al., 2021). However, an efficient image-based data augmentation technique for MIL is still an unmet need. We applied an inovative image augmentation strategy based on generating multiple dataloaders through different image transformations. The method showed increased performance compared to approaches without data augmentation. while the foundation model with proposed image augmentation consistently performed well, while ResNet and DenseNet showed variability, likely due to the inherent challenges of predicting aneuploidy from slide images and dataset imbalance. Combining augmentation strategies with the foundation model may ensure more robust performance across challenging tasks. Additionally, we implemented PseMix as a feature-level data augmentation method in our training. From the results of Appendix C, we found that PseMix augmentation improves the performance of MIL models in the presence of basic image augmentation. This experiment could further verify the adaptability of combinations of feature-based and image-based augmentations.

The top attention-scored patches provide crucial insights into the model's predictive features. Notably, DNA content abnormal patches showed enlarged, hyperchromatic nuclei with a loss of cell nucleus orientation and frequent mitotic figures. Traditional cell feature calculations for these patches revealed significant differences between abnormal and diploid samples in multiple features, including nucleus size and texture differences. The observed nuclear volume and changes in chromatin distribution align with known manifestations of increased chromosome content (Fischer, 2020; Chow et al., 2012). These findings emphasize the model's ability to explain relevant morphological disparities.

Despite the our method's inspiring performance, some limitations remain. Firstly, utilizing flow cytometry for the DNA content ground truth requires tissue disaggregation and destruction, restricting image analysis to adjacent tissue. Consequently, focal DNA abnormalities may lead to differing ploidy states in biopsy halves. Secondly, we argue non-BE regions may hinder the performance of abnormality prediction by weighting irrelevant regions, like dense inflammation. Future research will involve a Barrett's epithelium segmentation model to focus solely on relevant image patches.

To conclude, we showed the utility of a weakly supervised approach on feature prediction in a precancerous lesion, Barrett's esophagus. The novel image augmentation strategy along with utilization of a foundation model backbone showed high performance. Our model is efficient, capable of processing hundreds of samples in a tissue preserving setting, and thus an ideal adjunct to standard histologic evaluation. This classifier may facilitate molecular-based improvements in identifying both individuals at high risk for progression as well as those that may have already developed EAC.

## Acknowledgments

C.E. was supported by a Swiss National Science Foundation Postdoc Mobility fellowship (SNSF) (project number P500PM_214162). TGP was supported by NCI grants U2C-CA271902, R21CA2259687 and R01CA140657. WMG was supported by NCI grants UO1CA152756, and U2CCA271902.

We would like to thank Dr Luisa M Solis Soto, Jianling Zhou, and laboratory staff (Department of Translational Molecular Pathology, University of Texas MD Anderson Cancer Center) for their support in slide scanning. The authors acknowledge the support of the High Performance Computing for research facility at the University of Texas MD Anderson Cancer Center for providing computational resources that have contributed to the research results reported in this paper. The first illustration in Figure 1 Panel A was reproduced from Haidry RJ and Magee C. UEG Education 2018; 18: 12–14. (Haidry and Magee, 2018) The other illustrations in the panel were reproduced using BioRender's illustrations.

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

## Appendix A. Image augmentation transformations

```
import albumentations as A

transformations = A.Compose([
  A.HorizontalFlip(p=0.5),
  A.RandomBrightnessContrast(brightness_limit=0.2,
                       contrast_limit=0.2, p=0.5),
  A.GaussNoise(var_limit=(10.0, 50.0), mean=0,
                       always_apply=False, p=0.5),
  A.GaussianBlur(blur_limit=(3,7),
                       always_apply=False, p=0.5),
  A.ShiftScaleRotate(shift_limit=0,
                       scale_limit=0.1, rotate_limit=0, p=0.1),
  A.RandomRotate90(p=0.3),
  A.HueSaturationValue(p=0.2),
  A.RandomGamma(gamma_limit=(80, 120),
                       eps=None, always_apply=False, p=0.5)
])
```

## Appendix B. Augmentation experiment results

Table 2: Augmentation experiment results

| Experiment | Backbone | AUC (±95% CI) |
|---|---|---|
| Only Original | ResNet50 | **0.804 (0.75-0.859)** |
| | DenseNet121 | 0.431 (0.318-0.544) |
| | Foundation | 0.81 (0.583-1.037) |
| | | |
| Only Transformed | ResNet50 | 0.772 (0.754-0.79) |
| | DenseNet121 | 0.515 (0.464-0.566) |
| | Foundation | 0.459 (0.432-0.485) |
| | | |
| Original + Transformed | ResNet50 | 0.642 (0.581-0.703) |
| Concatenated | DenseNet121 | 0.416 (0.389-0.443) |
| | Foundation | 0.49 (0.468-0.512) |
| | | |
| Original + Transformed | ResNet50 | 0.763 (0.722-0.804) |
| Altering | DenseNet121 | **0.607 (0.425-0.788)** |
| | Foundation | **0.905 (0.879-0.932)** |

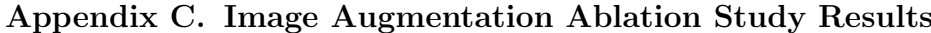

## Appendix C. Image Augmentation Ablation Study Results

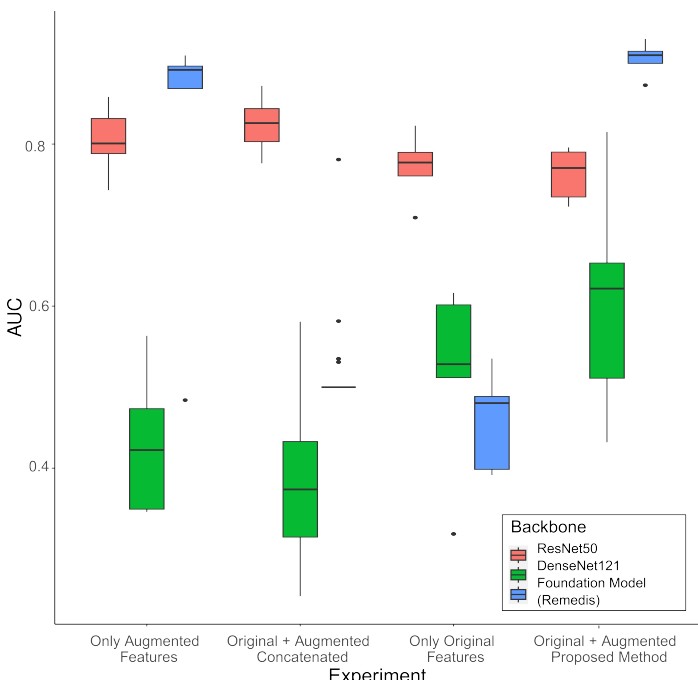

Figure 4: Augmentation experiment results. AUC from training without augmentation, simple concatenation of features, only augmented features and the proposed augmentation method.

## Appendix D. Data-level augmentation results on single split

Table 3: Data-level augmentation results on single split

| Backbone | No Augmentation | PseMix | PseMix + Image Augmentation | Image Augmentation |
|---|---|---|---|---|
| ResNet50 | **0.788** | 0.301 | 0.502 | 0.723 |
| DenseNet121 | 0.532 | 0.500 | 0.643 | **0.877** |
| Remedis 152x2 | 0.484 | 0.502 | 0.765 | **0.930** |

