# OpenReview forum: "Predicting DNA Content Abnormalities in Barrett’s Esophagus: A Weakly Supervised Learning Paradigm"
_MIDL.io/2024/Conference — MIDL 2024 Poster_

### Official Review · Reviewer_2fHh · 2024-02-25

**Confidence:** 4
**Preliminary Rating:** 3
**Final Rating:** 2.5

**Summary:**

This paper describes some experiments on the creation and validation of models for detecting aneuploidy (an important cancer biomarker) from standard H&E stained whole slide images (WSIs) taken from biopsies of Barretts oesophagus. The authors adapt the popular CLAM multiple instance learning model architecture in which pretrained feature extractors (three are tested in this paper) are run on each patch, and then a "bag" of patches is aggregated by learnable attention-based mechanisms. Furthermore the authors introduce an augmentation process by which augmented patch-level features are introduced during the training process.

**Strengths:**

The strength of this paper lies in its application. To my knowledge, this is the first paper to attempt to classify aneuploidy from whole slide images of Barrett's oesophagus. With a performance of AUROC=0.81, the performance of the best model is strong for this task. Furthermore, the authors link this to the clinical cancer outcome, showing that the prediction is not only possible with a reasonable degree of accuracy, but that the reasonably accurate prediction is accurate enough to have clinical utility.

Generally the paper is clear and well written.

**Weaknesses:**

The methods part of this paper is claimed as a key contribution but is considerably weaker than the application. The authors introduce a method for introducing augmentations into the training of a multiple instance learning approach. This is not necessarily straightforward for these models since the feature vectors are pre-computed as it is computationally too expensive to perform feature extraction again on each iteration of the MIL model training. In this paper, the approach is to precompute features from a limited number (4) of augmented versions of each tile, and then incorporate these into the training process. This appraoch has very limited novelty (being esentially the standard way to do augmentation in other types of model) and unfortunately is somewhat limited as it does not scale well to a greater number of augmetnations without taking up considerable storage to save the precomputed features. However, the authors show that an improvement can be seen with 4 versions, which is reasonable. The biggest problem however is that, while augmentation within MIL approaches is a problem ripe for further investigation, there are some other papers describing successful methods to do this (including xUP-MIL and ReMix, both cited in this manuscript), but the proposed method is not compared to these existing methods. This means it is impossible to evaluate the importance of this contribution.

A further weakness is that there are some aspects of the results that reduce my confidence in them. It seems very strange to me that the features extracted from the original images using a DenseNet model would give essentially a useless classifier (AUROC = 0.43) but features extracted in the same way from ResNet would give a *much* better classifier (AUROC=0.80), even though the two models were trained on the same dataset (ImageNet). This is counter to all my experience with these models, in which their performance is very similar. Similarly, I find the huge range between the AUROCs of the five cross-validation folds for the ResNet model (from < 0.5 to > 0.8) very troubling and suggestive of a significantly undersized dataset for this task or a deeper problem with the training process.

**Detailed Comments:**

- The authors state that "each network underwent modification to convert its output into a 1024 dimensional feature vector for every patch". Since these networks are pretrained, their architectures cannot straightforwardly be altered to change the size of the output feature vector. Therefore the authors should give specific details on how this step was achieved.
- A simple clarifying question: for a given augmented set of features, were the same augmentations applied to all patches in a WSI, or was a different random augmentation applied to each patch?
- Section 2.3 is presented without any context as to its relevance to the rest of the paper. Only at the end of the results section does the motivation for this analysis become clear. Its purpose should be clarified earlier in the paper.
- There is no reason for the line plot in Figure 2A. The ordering of the strategies along the x axis is arbitrary and has no meaning. A box and whisker plot would be more approrpriate here.

- It seems quite plausible to segment and classify nuclei, extract features from them, and then differentiate diploid from aneuploid cases on the basis of these features. This raises an obvious question of whether a simple "classical" classification approach based on these features may perform as well or perhaps better than a deep learning end-to-end approach.

Minor comments:
- The subpanels of Figure 2 are not explained. Which specific training strategy is represented in Figure 2B, for example? And which model's confusion matrix is shown in 2C? This is in the text but should be in the figure caption.
- What are the four categories on the x-axis of Figure 3D? E.g. what is meant by "diploid-diploid"? I would expect two categories, just "diploid" and "aneuploid".
- How many texture features were analyzed, and was a multiple hypothesis correction applied.
- There are lots of missing spaces in the text before parentheses and citations.

**Justification Of Final Rating:**

The inclusion of the PSEMix augmentation results as a baseline is a good addition and does much to address my comment about comparison to other established augmentation methods for MIL WSI models.

My comment about the feature vector size was unfortunately not addressed.

However, the new results are also highly erratic. For example, PSEMix reduced the performance of the ResNet50 model from 0.788 to 0.301 (an enormous drop in performance) while for the other models (which we would expect to behave quite similarly) it increased performance dramatically. This leads me to suspect that these models are trained and validated on datasets that are too small to have any confidence in the results presented in this paper. I therefore lean towards rejection.

**Justification Of The Preliminary Rating:**

This is a fairly strong applications paper with an important application and good results that will be of interest to the MIDL community, but the claimed technical contributions are weak and of limited value. This has led me to  give a borderline preliminary rating.

**Questions To Address In The Rebuttal:**

The claims of novelty of the proposed augmentation approach should be toned down, as should the conclusions about the effectiveness of the method as other published approaches have not been evaluated.

---

> ### Author Response · Authors · 2024-03-18
>
> R: Other Augmentation methods
> We appreciate the suggestion from the Reviewer. Due to time constraints, we evaluated our method against PseMix (Liu et al., IEEE TMI 2024), which is well-established in the literature. While we recognize the importance of feature-based augmentation methods, we believe image-level augmentations offer unique advantages, such as addressing staining and quality variations. Combining PseMix with image augmentation improved performance compared to no augmentation, although image-based augmentation alone showed superior results. Further fine-tuning may be needed for PseMix to be effective in challenging tasks like aneuploidy prediction. This experiment underscores the potential of combining feature-based and image-based augmentation methods in MIL models. We have updated the results and discussion sections accordingly.
>
> R: Why is ResNet50 better than DenseNet121? Why is the range huge?
> We appreciate the reviewer's feedback and concerns regarding the variability in performance between ResNet50 and DenseNet121 models.
> Upon investigation, we discovered an oversight in the training process of the DenseNet121 model related with initialisation of final layer during the initial submission. This adjustment improved the performance of the DenseNet-based models, resulting in performance slightly better than ResNet50.
> We appreciate the acknowledgment of the concern regarding the variability in AUROC values observed in the cross-validation folds for the ResNet model. Upon reflection, we recognize that this variability may indeed be attributed to potential issues with the dataset size, particularly the imbalance between the positive and negative classes.
>
> R: were the same augmentations applied to all patches in a WSI, or was a different random augmentation applied to each patch?
> We apologize for any confusion. Each tile in every slide underwent different augmentations with varying factors. The application of image transformations (such as blur, brightness, and zoom) was randomly determined for each image patch. Additionally, the factor for each image transformation was randomly selected within a specified range. Detailed probabilities and ranges for each transformation can be found in Appendix A.
> In addition to the original patch features, we generated four sets of features after applying image transformations. To ensure consistency across different backbones, we assigned predefined seeds for the generation of each set of feature vectors. By using different seeds for each set of augmented features, we ensured the utilization of different views of each patch for feature extraction, resulting in a variety of training data. This approach maintained diversity between sets within a training session while ensuring uniformity between backbones.
>
> R: Section 2.3 is presented without any context
> We are sorry for the oversight and appreciate the reviewer's suggestion. The cell feature classification analysis serves multiple purposes: (1) Understanding and explaining the histological features important for MIL model predictions, and (2) validating these features with known DNA content abnormality-related cell nucleus changes, such as enlarged and hyperchromatic nuclei. We have included a sentence in the introduction related to this part, and the relevant methods and results sections have been elaborated accordingly.
>
> R: What are the four categories on the x-axis of Figure 3D? E.g. what is meant by "diploid-diploid"? I would expect two categories, just "diploid" and "aneuploid".
> We are sorry for the confusion. The X-axis labels are showing the ground truth and predictions together. The part before underscore is the reference (flow analysis result) and the second part is the prediction. The figure was updated for clarity.
>
> R: How many texture features were analyzed, and was a multiple hypothesis correction applied.
> We thank to reviewer to draw our attention to the statistical calculation point.
> We calculated in total 24 features ( 6 shape, 5 intensity and 13 texture features).
> Formerly we did not apply any multiple hypothesis correction. After the reviewer’s comment, we applied Benjamini & Hochberg (1995) ("BH") p adjustment and the fig 3D was updated based on adjusted p values.
>
> R: simple "classical" classification approach based on cell features
> A: We thank to the reviewer for the suggestion. Although the cell features show significant difference on aneuploid samples, the analysis is depended on MIL classification. The features are calculated in the epithelial cells in top-attention scored patches. Given that aneuploidy is often a local event, it is un expected that analysis of whole slides and training a basic classifier provides similar performance.
>
> R:Fig2A
> The figure was updated and moved to appendix.
>
> R: There are lots of missing spaces in the text before parentheses and citations.
> We apologise for the grammar mistakes. The errors were corrected.

---

### Official Review · Reviewer_wUgz · 2024-02-27

**Confidence:** 5
**Preliminary Rating:** 3
**Recommendation:** Poster
**Final Rating:** 3.5

**Summary:**

This work is devoted to detecting DNA abnormalities in Barrett's esophagus. These abnormalities are implicated in the progression to cancer. For that matter, the authors are using histology images. Using the MIL paradigm, detection is using bag of tiles with annotated slides. These tiles are encoded through a feature extractor network. In this work, the authors propose a novel technique alternating at training between features from original slides and features extracted after augmentation of these tiles. A cohort from Seattle is used from which an independent test set is extracted and uses a different slide scanner. Satisfactory results are obtained on the binary classification (diploid vs aneuploid). To investigate the model's predictive features, top attention-scored patches and their cell morphological features are analysed.

**Strengths:**

The main strengths of the paper are:
- interesting clinical question, convincing results are obtained with the proposed approach. As the test cohort uses a different scanner, it shows promising generalization,
- the simple yet effective idea for the augmentation strategy is well described and could easily be applied in contexts,
- results are carefully analysed and discussed, including a correlation between DNA abnormalities and cancer progression,
- limitation of the work are honestly described and its perspectives are promising.

**Weaknesses:**

Besides the limitation on ground truth that is identified in the paper, the main weakness is the statistical power of the results. Although the number of patients is significant, there seems to be a significant variability (e.g. on AUC as shown in Fig 2) between folds (not for the combination of the foundation model and the strategy).

One could also have investigated different augmentation strategy to really show that the proposed approach is optimal. As each patch has 5 versions of its features (original + 4 extractions after augmentation), the dataloader could have taken one of its version randomly on the fly.

**Detailed Comments:**

Besides the main strengths and weaknesses of the work listed previously:
- paper is well-written with a careful description of the training setup / hyperparameters / software which eases reproducibility
- I do not understand why on page 4, the first cohort is split into training (85%) / validation (15%) while on page 6, the authors write that "all training sessions utilized 5-folds of training/validation splits"? It is a leftover from a previous version.
- on table 1, the balanced accuracy is fairly low for ResNet and DenseNet backbones. Could the threshold be adjusted to improve this score (I suppose .5 is used but it is not written in the document as it should).

**Justification Of Final Rating:**

The paper has improved but many typos need to be corrected. I still have questions about the robustness of the approach considering the large difference in results between backbones/augmentation strategies. It should be discussed a bit in the document.

**Justification Of The Preliminary Rating:**

This work is definitely interesting and the study properly conducted. The paper is well-written and pleasant to read. The presented results are satisfactory yet given the simplicity of the augmentation strategy it could been a bit more investigated / validated (esp for the generalization ability of the approach).

**Questions To Address In The Rebuttal:**

The main questions to address in the rebuttal would be on the evaluation of the generalization of the approach (e.g. with a discussion of the large error bars in AUC on Fig 2) and the validation of the alternation strategy.

**Special Issue:**

No

---

> ### Author Response · Authors · 2024-03-18
>
> One could also have investigated different augmentation strategy
> We thank the reviewer for the suggestion. We implemented PseMix (Liu et al., IEEE TMI 2024) data-augmentation method to the study. The reason for selecting PseMix was its comprehensive demonstration of robustness in published literature.
> However, we propose image-level augmentation as an alternative to feature-level data augmentation (e.g., PseMix) because they offer different strengths and weaknesses. Image-level augmentations provide variation at the image level for possible staining, scanner, or slide quality-related variations, such as color, brightness, noise, blur, saturation, etc., which differs from Mixup and pseudobag-based feature-level data augmentation methods.
> Combining PseMix and image augmentation resulted in improved performance compared to the no augmentation setting. Yet, image-based augmentation alone showed superior performance compared to PseMix. While the authors demonstrated the robustness of the PseMix method in tumor classification tasks (Liu et al., IEEE TMI 2024), fine-tuning may be required for the clustering, masking, phenotype-stratified sampling, and mixing steps to work effectively in a more challenging setting such as DNA content abnormality (aneuploidy) prediction in small esophagus endoscopical biopsies. This experiment highlights the effectiveness and potential of combining feature-based and image-based augmentation methods in MIL models in computational pathology.
> We have updated the results and discussion sections accordingly.
>
> Could the threshold be adjusted to improve this score
> We apologize for the oversight regarding the threshold value for WSI-level classification. The default threshold of 0.5 was indeed used, but it was not explicitly stated in the document. Following your recommendation, we calculated the optimal threshold using the ROC curve and the TPR-(1-FPR) formula based on probabilities of the training dataset samples. As a result, the balanced accuracies for DenseNet and ResNet were noticeably improved. We have updated the manuscript text and tables to reflect this methodology and the results. The thresholds for the test set were adjusted from 0.41, 0.13, and 0.85 to 0.67, 0.50, and 0.83 for ResNet, DenseNet, and the foundation model, respectively. Table 1 has been updated accordingly. We appreciate the insightful recommendation from the reviewer.
>
> I do not understand why on page 4, the first cohort is split into training (85%) / validation (15%) while on page 6, the authors write that "all training sessions utilized 5-folds of training/validation splits"? It is a leftover from a previous version.
> We apologize for the confusion. we applied 5 iterations of Monte Carlo cross-validation for the training. Therefore, there is no leftovers. We have updated the relevant section in the methods to clarify this discrepancy. Thank you for bringing this to our attention.

---

> > ### Comment · Reviewer_wUgz · 2024-03-25
> >
> > The paper has clearly been improved in this revision. Yet I am not completely convinced by the results. One may question the robustness of the approach esp when looking at Table 2 in appendix B and the difference between backbones/strategy combinations.
> > There are still many typos in the paper: Model size small on page 4, "stuy", "ReNet" on page 6, "transfromations" in appendix...

---

> > > ### Author Response · Authors · 2024-03-27
> > >
> > > We appreciate the reviewer's feedback and their acknowledgment of the improvements made in this revision.
> > > We acknowledge the concerns of the reviewer. Predicting DNA content abnormality from WSIs indeed presents a challenging task. Our study aimed to address this challenge by exploring various backbone models and augmentation strategies. The main output model of the study, foundation model as the feature extraction backbone combined with image-based augmentation, demonstrates a robust performance at decent consistency.
> > > While DenseNet and ResNet have demonstrated success in tasks such as tumor classification, achieving robust and consistent performance in a challenging task like DNA content abnormality prediction may require alternative solutions. We have duly noted the limitations of our study regarding the DenseNet and ResNet experiments in the limitations section of the manuscript.
> > > We apologize for any oversights in grammar and typos. The manuscript has been thoroughly checked and these errors have been corrected. Additionally, we have amended the figure legends to enhance the overall clarity of the document.

---

### Official Review · Reviewer_iKuQ · 2024-02-28

**Confidence:** 4
**Preliminary Rating:** 3
**Final Rating:** 3.5

**Summary:**

The study proposes a weakly supervised multi-instance learning (MIL) method to predict ploidy status in Barrett's esophagus (BE) using routine histology images. The top-performing model achieved a high accuracy of 0.93 AUC and 0.75 balanced accuracy in predicting aneuploidy in test cohort biopsies, showing correlation with BE progression to esophageal adenocarcinoma (EAC) and suggesting potential for identifying high-risk individuals.

**Strengths:**

It is the first study tackling the prediction of DNA content status based on histology images of BE.
Good AUC results are obtained. It is also interesting to show the correlation with cancer outcomes. The analysis of morphologic features of high attention patches is also interesting. On the same line, you could also consider a concept attribution analysis as in Graziani, Mara, et al. "Concept attribution: Explaining CNN decisions to physicians." Computers in biology and medicine 123 (2020): 103865.

**Weaknesses:**

The motivation for the augmentation method is a bit weak (primarily training with original images while leveraging the benefits of image augmentation to enhance overall performance and generality). It is shown empirically to work on this dataset, but it would be good to validate on other sets and compare with other approaches.

The method is compared only with a baseline without any augmentation. It is not the best comparison. It should be compared with existing augmentation methods, e.g. Liu et al., 2023; Gadermayr et al., 2023; Yang et al., 2022.

There are some incoherencies I do not understand (see detailed comments).

**Detailed Comments:**

1) Incoherencies:

In the abstract, it reads all patients n=117. But I read 34+89 patients in 2.1.

I do not understand the results in the abstract and rest of the paper later. 0.91 AUC is mentioned (again in Appendix B), then 0.93 (again e.g. in Table 1).

In the abstract, you mention 2 backbone models (DenseNet and REMEDIS), but ResNet is also used in the Results.

2) Annotations are at the WSI level. If a correlation is found between aneuploidy and EAC progression, why not also trying to predict directly EAC?

3) I am not sure all the mentioned packages and versions are necessary. It would be better to provide the code and requirements on github.

4) Some typos to fix, e.g. "as the presence aneuploidy", "necessitate use of precious", "provides the model is primarily" etc.

**Justification Of Final Rating:**

My final recommendation is borderline accept because the authors clarified multiple points, improved the manuscript (based on all reviewers comments). They also added a comparison with PseMix, and a combination of feature- and image-level augmentation. The sentences added to the manuscript must, however, be checked for grammar and typos.

**Justification Of The Preliminary Rating:**

The paper is relatively clear, well written and proposes a novel application of DNA content status prediction from histology images of BE.
It lacks comparison with other augmentation methods and stronger motivation and/or vlidation of on other tasks/datasets of the proposed augmentation strategy.

**Questions To Address In The Rebuttal:**

Please fix or explain the mentioned incoherencies.
A comparison with other augmentation methods is also needed.

---

> ### Author Response · Authors · 2024-03-18
> **Thank you for the constructive feedback and suggestions provided by the reviewer. We have carefully addressed each point raised and updated the manuscript accordingly. Below are our responses to each of the reviewer's comments:**
>
> A comparison with other augmentation methods is also needed.
>
> We appreciate the reviewer's suggestion to compare our image augmentation method with other data augmentation methods developed for MIL training. Due to time constraints for revision, we opted to select one approach and test it against our results. Among the available options, we chose PseMix (Liu et al., IEEE TMI 2024) due to its comprehensive demonstration of robustness in the published literature.
> However, we propose image-level augmentation as an alternative to feature-level data augmentation (e.g., PseMix) because they offer different strengths and weaknesses. Image-level augmentations provide variation at the image level for possible staining, scanner, or slide quality-related variations, such as color, brightness, noise, blur, saturation, etc., which differs from Mixup and pseudobag-based feature-level data augmentation methods.
> Combining PseMix and image augmentation resulted in improved performance compared to the no augmentation setting. Yet, image-based augmentation alone showed superior performance compared to PseMix. While the authors demonstrated the robustness of the PseMix method in tumor classification tasks (Liu et al., IEEE TMI 2024), fine-tuning may be required to work effectively in a more challenging setting such as DNA content abnormality (aneuploidy) prediction in small esophagus endoscopical biopsies. This experiment highlights the effectiveness and potential of combining feature-based and image-based augmentation methods in MIL models in computational pathology.
> We have updated the results and discussion sections accordingly.
>
>
> I do not understand the results in the abstract and rest of the paper later. 0.91 AUC is mentioned (again in Appendix B), then 0.93 (again e.g. in Table 1).
> We apologize for any confusion regarding our performance metrics. The performance metrics were reported in two forms:
> 1- The average of multiple Monte Carlo cross-validation training/validation splits (n=5), as indicated in Table Appendix B, and the second paragraph of the abstract.
> 2- The top AUC score from a specific training/validation split, as depicted in Figure 2A-B, Table 1, and the third paragraph of the abstract. The model utilizing this split and the foundation for feature extraction was selected for further analysis, including cell morphology correlation analysis.
>
>
> In the abstract, it reads all patients n=117. But I read 34+89 patients in 2.1.
> We apologise for the mistake. The correct patient count is 123. The number in the abstract was wrong. It is corrected.
> In the abstract, you mention 2 backbone models (DenseNet and REMEDIS), but ResNet is also used in the Results.
> Thank you for noticing the missing part. We added the average AUC of features based ResNet50 in the abstract.
> Annotations are at the WSI level. If a correlation is found between aneuploidy and EAC progression, why not also trying to predict directly EAC?
> We are grateful to the reviewer for raising an important point regarding the direct prediction of esophageal adenocarcinoma (EAC) and its potential clinical implications. While our current focus centers on unraveling the biological significance of aneuploidy in Barrett Esophagus, we acknowledge the merit of exploring direct prediction models for EAC. Additionally, aneuploidy is the most strongly correlated biological change in BE which can be predictive for EAC.
> In our study, our primary objective is to delve into the underlying biological mechanisms of molecular alterations, specifically aneuploidy, within Barrett Esophagus and its association with cancer progression. While predicting EAC directly is certainly an intriguing prospect, our focus lies in elucidating the significance of aneuploidy in this context. By training our model to predict aneuploidy, we not only provide valuable molecular and clinical prognostic insights but also pave the way for subsequent analyses. We think, investigation of the provenly correlating middle steps in the cancer progression can provide valuable insight and soil for deeper understanding of the progression.
>
>
> I am not sure all the mentioned packages and versions are necessary. It would be better to provide the code and requirements on github.
> We thank the reviewer for the suggestion to simplify the manuscript. We have made the necessary changes in the methods section accordingly. The list of used packages and the code will be shared upon publication.
>
>
> Some typos to fix, e.g. "as the presence aneuploidy", "necessitate use of precious", "provides the model is primarily" etc.
> We thank to the reviewer for bringing those typos to our attention. We have corrected them accordingly.

---

> > ### Comment · Reviewer_iKuQ · 2024-03-26
> >
> > Thank you for your response and modifications.
> > The sentences added to the manuscript must, however, be checked for grammar and typos.

---

> > > ### Author Response · Authors · 2024-03-27
> > >
> > > We apologize for any oversights in grammar and typos. The manuscript has been thoroughly checked and these errors have been corrected. Additionally, we have amended the figure legends to enhance the overall clarity of the document.

---

### Meta-Review · Area_Chair_ThEW · 2024-04-06

**Recommendation:** Accept (Poster)
**Confidence:** 3

**Metareview:**

Reviewers initially considered this work has some value but needed some improvements, which were mostly done during the rebuttal period. As a result, two reviewers lean towards acceptance, whereas one leans towards rejection, indicating in their final comment that these models might be trained and validated on datasets that are too small, which is something that must then be addressed ands discussed in the paper.

---

### Decision · Program_Chairs · 2024-04-05

Accept (Poster)